# Studying the Influence of Salt Concentrations on Betalain and Selected Physical and Chemical Properties in the Lactic Acid Fermentation Process of Red Beetroot

**DOI:** 10.3390/molecules29204803

**Published:** 2024-10-11

**Authors:** Emilia Janiszewska-Turak, Anna Wierzbicka, Katarzyna Rybak, Katarzyna Pobiega, Alicja Synowiec, Łukasz Woźniak, Urszula Trych, Andrzej Krzykowski, Anna Gramza-Michałowska

**Affiliations:** 1Department of Food Engineering and Process Management, Institute of Food Sciences, Warsaw University of Life Sciences—SGGW, 159C Nowoursynowska St., 02-787 Warsaw, Poland; annawierzbicka19@wp.pl (A.W.); katarzyna_rybak@sggw.edu.pl (K.R.); 2Department of Food Biotechnology and Microbiology, Institute of Food Sciences, Warsaw University of Life Sciences—SGGW, 159C Nowoursynowska St., 02-787 Warsaw, Poland; katarzyna_pobiega@sggw.edu.pl (K.P.); alicja_synowiec@sggw.edu.pl (A.S.); 3Department of Food Safety and Chemical Analysis, Institute of Agricultural and Food Biotechnology, 36 Rakowiecka Street, 02-532 Warsaw, Poland; lukasz.wozniak@ibprs.pl; 4Department of Fruit and Vegetable Product Technology, Institute of Agricultural and Food Biotechnology, 36 Rakowiecka Street, 02-532 Warsaw, Poland; urszula.trych@ibprs.pl; 5Department of Thermal Technology and Food Process Engineering, University of Life Sciences in Lublin, 31 Głęboka St., 20-612 Lublin, Poland; andrzej.krzykowski@up.lublin.pl; 6Department of Gastronomy Science and Functional Foods, Faculty of Food Science and Nutrition, Poznan University of Life Sciences, Wojska Polskiego 31, 60-624 Poznan, Poland; anna.gramza@up.poznan.pl

**Keywords:** lactic acid fermentation, beetroot, *Beta vulgaris* L., betalains, salt addition

## Abstract

This study emphasizes the significance of optimizing salt content during the fermentation of red beetroot to produce healthier and high-quality fermented products. It investigates the impact of different salt levels on fermentation, analyzing various parameters such as pH levels, dry matter content, total acidity, salt content, color changes, pigment content, and lactic acid bacteria count. This study identifies the most favorable salt concentration for bacterial growth during fermentation and storage as 2–3%. It was evaluated that salt levels fluctuated significantly during fermentation, with nearly 50% of the added salt absorbed by the beetroot tissues, mainly when lower salt concentrations were used. The fermentation process had a negative effect on the content of betalain pigments, as well as yellow pigments, including vulgaxanthin-I. It was also found that fermentation and storage affected the proportions of red pigments, with betacyanins proving to be more stable than betaxanthins, and that salt addition affected negatively pH and total acidity while causing an increase in yellow color. The pH was negatively correlated with the duration of the process, the amount of red pigment, and bacterial count. The results indicate that lower salt levels can lead to favorable physicochemical and microbiological parameters, allowing for the production of fermented red beetroot with reduced salt content without compromising quality.

## 1. Introduction

In recent years, fermented products, among other vegetables, have grown in popularity and are in high demand. This trend is related to the public’s growing interest in healthy lifestyles and nutritional awareness [1,2,3]. The presence of lactic acid bacteria in fermented products has been found to benefit human health. While dairy products are commonly fermented [4], some consumer groups exclude such animal-derived products. As a result, plant-fermented products are gaining traction, especially among lactose-intolerant individuals and vegans [5,6].

Of the frequently fermented vegetables, cabbage, cucumbers, olives, beetroot, peppers, broccoli, cauliflower, and tomatoes are well known. In addition, compound products such as kimchi, miso, and kombucha, which have traditional roots in Asia, have gained popularity in recent years [1,7]. Fermenting individual vegetables or mixtures of vegetables is performed to achieve new textural and structural properties, enhance health-promoting qualities, and prolong shelf life. This process not only preserves the active components of the vegetables but also enriches them with new ones, contributing to the growing popularity of fermentation [8,9].

Beetroot, scientifically known as *Beta vulgaris* L., typically presents as a red vegetable, although variations in white, yellow, and red–white with distinct striations are also available [10,11]. Owing to its rich content of active substances, such as polyphenol pigments, beetroot has become an invaluable commodity in the vegetable industry. Beetroot is particularly rich in betalain pigments, which give it its characteristic color and antioxidant properties [12,13,14,15]. In addition, beetroot contains numerous vitamins and minerals, including folate, polyphenols, carotenoids, and fiber [2,11,12,16].

Lactic acid bacteria, during their growth, produce beneficial metabolites such as lactic acid, which primarily contributes to fermentation, and bacteriocins, natural preservatives for fermented products [17,18]. Certain lactic acid bacteria have been identified for their probiotic potential. The term refers to “live microorganisms that, when administered in adequate amounts, provide health benefits to the host” [19]. The beneficial effects of lactic acid bacteria stem from their ability to directly influence the balance of intestinal microflora and inhibit pathogens [6,20,21,22].

The fermentation of vegetables offers a means of preservation and a range of other benefits [23]. It allows fermentation to produce a new product with an extended shelf life due to the reduced pH of the resulting lactic acid. It enhances the availability of nutrients and pigments, increasing the availability of ingredients and facilitating better absorption of antioxidants, among other advantages. Additionally, the fermentation process alters the sensory properties, thereby improving digestibility. In the case of fermented beet, the fermentation process minimizes the perception of the characteristic ‘earthy taste’ caused by geosmin [9,24].

During fermentation, it is crucial to maintain the right conditions, which is why starter cultures appropriately matched to the raw material are selected. Commonly used cultures for vegetable fermentation include *Lactiplantibacillus plantarum*, *Levilactobacillus brevis*, *Lactiplantibacillus pentosus*, and *Limosilactobacillus fermentum*. For these strains, it is essential to specify the process conditions: the temperature should typically range from 25 °C to 30 °C, and the fermentation time should be long enough to obtain the highest number of lactic acid bacteria in the final product, usually 7 to 14 days [1,21,25]. The concentration of salt in the brine is a critical factor; adding NaCl at a level of about 1–5% positively affects the growth of LAB bacteria and inhibits the development of undesirable microflora. Insufficient levels will fail to inhibit the growth of pathogenic microflora, while excessive levels will hinder LAB growth. Moreover, salt influences sap secretion by plant cells, facilitating the bacteria’s access to nutrients [1,26]. In the literature, we can find information that levels of 0–1.5 or even 2% do not significantly impact the bacteria growth, which means all microorganisms can grow LAB and others treated as pathogenic or inappropriate microbiota too (yeasts, molds). Higher levels than 2–3% can reduce growth for *E. coli* or *P. fluorescens* [27,28]. A level above 5% is treated as the one for halophilic bacteria growth [29], while LAB strains can reduce even up to 68% of bacteria [30]. Bacteria respond to salt stress by regulating cytosolic pools of organic solutes to achieve osmotic equilibrium. Microorganisms have developed two effective strategies for acclimating to changing salt concentrations: the “salt-in-strategy” and the “salt-out-strategy” [27,28,31].

Salt is an essential nutrient for the body, but excessive consumption can lead to health issues. Processed foods often contain high levels of salt, contributing to overconsumption. The World Health Organization (WHO) advises limiting daily salt intake. Minor reductions in the salt content of products do not compromise taste and can help to decrease overall salt consumption [1,26,32,33,34,35]. The excessive use of salt in food processing adversely affects the environment, leading to high salinity in wastewater and increased treatment costs [36]. According to a study by Zhang et al. [37], lower salt content promotes the growth of lactic acid bacteria in Chinese paocai. Similarly, research by Yang et al. [38] indicates that a 0.5% salt content significantly enhances the maturation and quality of sauerkraut. Other authors mentioned the existence of off-flavors and the softening of the texture of fermented products [36].

Reducing salt in fermented vegetable products is essential to lower overall salt intake. Determining the optimal amount of salt for fermentation is crucial for producing sustainable, high-quality products. A study was conducted to investigate the effects of various salt levels (ranging from 0 to 6%) on the properties of fermented red beetroot, including pH, dry matter, acidity, color, betalain content, and bacteria counts during fermentation and storage. The reference conditions were set up with a salt level of 2%, the most commonly used for vegetables. The research hypothesis was that different salt content could be used for beetroot fermentation without altering the process results. The second hypothesis was that a different salt level would facilitate bacteria growth without causing changes in betalain.

## 2. Results and Discussion

### 2.1. Microbiology

This research aimed to determine the levels of lactic acid bacteria in red beetroot during fermentation and storage. Initially, the lactic acid bacteria count in the beetroot was measured at 4.88 ± 0.14 log CFU/mL right after the inoculum was added, possibly including natural LAB contaminants. By the fourth day of fermentation, there was a significant increase in bacterial counts, regardless of salt addition. However, for beetroot fermentation with brine containing 0%, 2%, and 4% salt, the increase ceased on the eighth day (Figure 1). Notably, the higher bacteria counts observed from the fourth to the eighth day for salt additions from 0 to 5% could be attributed to the logarithmic phase of LAB growth, indicating that fermentation could be completed after this period to preserve the maximum number of probiotic bacteria. Six percent salt in the brine decreased bacterial count after the fourth day, resulting in the lowest LAB values (below 8 logs CFU/mL) throughout the fermentation process. This decrease could be attributed to the osmotic stress caused by the salt level in the brine. Additionally, lactic acid bacteria can adapt to osmotic stress by employing strategies such as modifying their proteome profile, including upregulating proteins involved in cell wall metabolism and oxidative stress tolerance [30]. However, these adaptations were not observed in our study during eleven days of fermentation, all tested samples reached a stable point, and the bacteria began to decline, except for the 6% salt fermentation, where the decrease was apparent after the fourth day. Additionally, the decline was faster for samples with salt addition ranging from 4% to 6%. Following storage, the bacterial count decreased across all samples, with the lowest decrease for samples with 2% (16.6%) and 3% (15.7%) salt additions compared to the highest values observed on the fourth day. The bacteria thrived best in 2% and 3% salt concentrations, even after storage. The decrease in bacteria count on the final day of testing suggests that they may have entered the death phase. It is essential to monitor this process carefully to intervene at the right time and preserve the maximum number of beneficial lactic acid bacteria.

In all the samples, no signs of spoilage were detected. There was no visible presence of yeasts and molds. Additionally, the taste and smell were typical for fermented vegetables. When higher salt levels were added to the brine, a pronounced salty taste was observed. No visible changes in the structure and texture of the beetroot cubes were observed, for our other studies on yellow beetroot fermentation, similar observations for structure/texture were made [39].

The salt addition could affect the diversity of the microbial community during the fermentation. Previous research has indicated that the genus *Lactobacillus* is less tolerant to salt than other LABs, but the mechanisms underlying this phenomenon have not been thoroughly investigated [40]. Salt has been demonstrated to enhance the safety of fermented products by reducing water activity and effectively inhibiting the growth of spoilage and pathogenic microorganisms that may be present on vegetable surfaces. However, insufficient salt levels can decrease the firmness of the raw materials [41]. Microorganisms use salt in low concentrations for their activity, while higher concentrations lead to plasmolysis and, ultimately, the death of the microorganism. Changes in osmolarity have also been shown to alter various internal cellular processes, causing microbial death. In vegetable fermentation, plasmolysis can also cause the growth of lactic acid bacteria by releasing nutrients in plant cells [33,42]. A study by Casciano et al. [43] found that a blend of LAB bacteria resulted in a 6.07 log CFU/mL count after fermentation in 6% brine, consistent with previous laboratory research. The study suggests that reducing the salt content to a maximum of 3% leads to higher concentrations of lactic acid bacteria. Similarly, a study by Janiszewska-Turak et al. [44] on fermented red beetroots in 2% brine demonstrated a 7.62 log CFU/mL output, closely resembling the results observed in brines containing 1 to 3% red beets. Another study by Lamba et al. [45] examined the addition of salt (5%, 8%, and 10%) to the fermented Kanji drink and observed an exponential increase in fermentation between the 22nd and 30th day, depending on the amount of salt added. In the case of fermented peppers, it was found that the presence of lactic acid bacteria (LAB) was significantly reduced when 8% and 10% salt were used, as compared to 2–6% salt addition [46]. Studies have shown that the optimal salt concentration for pickled radish is 2.5%. However, in addition to salt concentration, lactose, MgSO_4_ + MnSO_4_, and mustard levels are also crucial [42].

### 2.2. Physical Properties of Fermented Beetroot

#### 2.2.1. pH and Total Acidity

After four days, the pH levels of the red beetroot brine samples were in the range of 3.45 to 3.77. By the 11th day, this range narrowed to 3.32 to 3.51 (Figure 2). It was observed that the 3% and 6% salt additions expedited the process, while 0% and 1% salt additions resulted in a slower process. A consistent pH level of 3.5 on the final day indicates stability, and a slight pH increase during the latter stages of fermentation might be attributed to yeast consumption of lactic acid. Following a 60-day storage period, the pH was approximately 3.2–3.3.

After re-evaluating the hypothesis regarding the effect of salt addition on pH, it was determined that the theory could be rejected. Adding salt to the brine did not impact the pH during the subsequent days of the fermentation process. However, statistically lower pH values were observed for salt concentrations of 3 and 4 on the final day of the process (day 11) and after the storage period. A pH below 4 indicates food safety, as it inhibits the growth of pathogenic microflora. Choińska et al. [47] conducted a study on the fermentation of beetroot using various LAB strains and noticed a decrease in pH values from 6.08 to 3.61–3.89 for the Woodan cultivar after seven days. Different salt additions (0.5, 1.5, and 5%) were also determined in the fermentation of cucumbers by Świder et al. [48], and the same behavior with decreasing the pH was observed.

When analyzing the data for total acidity, it was observed that there was an increase in values until the 11th day of the fermentation (Figure 2). After the storage period, the value remained almost the same as on the last tested day of fermentation. Hence, it can be concluded that the primary fermentation process ended after 11 days. The salt content did not influence the behavior of total acidity. However, a small increase was observed for samples with a 6% salt addition. This behavior is attributed to bacterial growth in those samples, with the lowest LAB growth.

Lamba et al. [45] observed a similar trend for the total acidity of fermented Kanji (beetroot and black carrot juice) with 8% salt addition. After about 4–6 days of fermentation, the total acidity values stabilized at 0.63–0.68 for twelve days and remained stable until the end of the storage. The pH values decreased during fermentation and reached 3.2 by the end.

#### 2.2.2. Dry Matter and Salt Content

The initial dry matter content of fresh red beetroots was 14.09% (Table 1) and the highest among all the samples. Throughout the fermentation process, the dry matter values decreased regardless of the salt content in the brine. This decrease could be attributed to the release of water-soluble substances into the brine and the utilization of sugars by microorganisms during fermentation [1]. The most significant decrease was observed in samples with 0% and 1% salt in the brine, amounting to approximately 40% compared to the fresh beetroot. However, increased salt content led to a statistically significant increase in dry matter in the samples. Following the initial days of fermentation, the dry matter content remained stable, with no fluctuations observed in samples with the same salt percentage. This behavior is comparable to osmotic dehydration, wherein water is extracted from the product and transferred into the osmotic solution. In contrast, osmotically active solutes are simultaneously transferred from the solution into the food product [49].

Furthermore, lactic acid bacteria use sugars to produce lactic acid during fermentation, decreasing dry matter. This reduction is associated with the catabolic breakdown performed by the bacteria [50]. Additionally, salt, as a component of the fermentation process, prompts the release of nutrients from beetroot cells through osmosis. The water-soluble nutrients serve as nourishment for lactic acid-producing bacteria [51].

The data in Table 1 illustrate that red beetroot undergoes substantial changes in salt levels at the onset of fermentation. After four days, samples without added salt (0%) in brine nearly matched the untreated beetroot’s salt levels, indicating natural NaCl presence. As the salt concentration in subsequent brines increased, the samples demonstrated higher salt levels. By the fourth day, except for 0% brine, the samples contained lower salt amounts than the brine’s added quantity. The diverse brine concentrations exhibited significant variations in values, with a significance level of α = 0.005. Notably, almost 50% of the added salt was transferred to the beetroot tissues, with higher percentages observed for lower salt addition (1 to 3%). According to Dimakopoulou-Papazoglou et al. [49] and Ahmed et al. [52], the salt present in brines acted through osmosis on the plant tissues, leading to salt diffusion into the tissue and from the tissue into the water-soluble substances in the brine, causing cell sap mixing with salt and brine with cell sap. The observed fluctuations in NaCl content during fermentation can be attributed to the pursuit of equilibrium in salt concentration between the brine and plant tissue.

#### 2.2.3. Color of the Fermented Beetroots

The color analysis results are detailed in Table 1. The parameter dE denotes the color variation between the fermenting beetroot samples and the raw beetroot reference sample. The distinct purple–red color of red beetroot is attributed to the prevalence of betaxanthins, particularly betanin, and yellow betacyanins, specifically vulgaxanthin-I. Furthermore, the color is influenced by the proportion of these pigments [53,54]. The color coefficients for the fresh beetroot sample were as follows: L* = 11.46 ± 0.62, a* = 16.32 ± 0.4, b* = 3.61 ± 0.27, which differed from the values reported by Fijałkowska et al. [55] (L* = 20.3, a* = 21, b* = 5.6) and our previous study on the same cultivar (L* = 13.1, a* = 17.7, b* = 1.9) [56]. While the same cultivar was used in the above research, the variations are associated with the harvest timing, encompassing factors such as the weather conditions during the beetroot’s growth cycle and other soil-related determinants. This is why the observed dE values for different authors can differ from ours.

The brightness of fermented beetroots increased throughout the fermentation process for each salt addition group until the 11th day, while, after storage, this coefficient decreased (Table 1). The highest values of color coefficients a* and b* were achieved with higher salt concentrations in the brine (ranging from 3 to 6%); for color coefficient a*, the values also declined during the fermentation process. After storage, a reduction in the color coefficient a*, corresponding to the red color, was observed.

As the duration of fermentation increased, the sample exhibited higher lightness, accompanied by an increase in lightness values. This phenomenon could be attributed to the escalating water content within the samples, leading to the dispersion of betalains into the brine. Furthermore, the values decreased again after storage in samples with 0–3% salt content. The most significant changes in the color factor were observed in the yellow–blue color factor (b*) in samples with a salt content of 0–2% in brine; regardless of the day of the process, the values were negative—indicating a shift towards yellow. No clear trends were observed in the color coefficients a* and L*.

According to the data in Table 1, most beetroot dE values ranged from 5 to 12, except day 6, where a 6% salt addition led to significantly larger color changes than fresh beetroots. Although no specific trend was observed, all samples showed notable color changes with values exceeding 2, even noticeable to less experienced consumers, as noted by Tiwari et al. [57]. The most significant changes were in samples placed in 6% brine after fermentation. Those variations are related to the change in the color coefficient during the process.

The conducted tests suggest that the fermentation process does indeed impact color changes. It is possible to minimize the loss of visual characteristics by choosing appropriate parameters, such as salt content. Several factors influence the degree of color change throughout fermentation, with salt content playing a pivotal role. Color alterations, especially in lightness, may occur due to water, salt, and pigment release during fermentation. The release of pigments from the beetroot into the brine during fermentation could also contribute to these changes. A higher salt content leads to a more pronounced exchange of constituents between the brine and plant tissue, potentially affecting color changes throughout fermentation.

### 2.3. Pigment Content

Based on our research, the levels of red (betacyanin) and yellow (betaxanthin) pigments in fresh beetroot cubes were measured at 210.35 ± 16.5 mg betanin/100 g dry matter and 115.23 ± 11.06 mg vulgaxanthin-I/100 g dry matter, respectively (Figure 3). These values can vary depending on cultivar, harvest time, and geographical region. Betacyanin accounted for 64.6% of all pigments, aligning with similar findings presented in a prior study by Barbu et al. [58], which reported a value of 69.5%. Furthermore, Nirmal et al. [53] noted that the ratio of betacyanin to betaxanthin could fall within the salt range of 1 to 3, which is the same as our results.

During the initial stages of the beetroot fermentation process with 5% salt addition, the highest levels of red pigment were observed, ranging from 240 to 270 mg betanin/100 d.m. These levels were 6 to 22% higher than the betanin content in raw beets. Two distinct trends were noticed in the early days of the fermentation process: beetroot fermented in 2% and 5% salt displayed an increase in pigment content. At the same time, the remaining samples exhibited a decreasing trend. However, that fluctuation was not deemed statistically significant. The variation in pigment content could be attributed to uneven pigment distribution within the beetroot tissue samples used for testing. Since betalains are water-soluble pigments, there is a potential for the pigment to migrate into the brine [14]. Despite the salt addition, a decline in red pigment content was noticed after storage. Similar patterns were identified by Thippeswamy et al. [59] in the storage of nonfermented beetroot extract with varying salt levels on the first day. Their conclusion highlighted that ideal betalain content and antioxidant activity were achieved with 1–3% salt, a common occurrence in fermented vegetables. Nevertheless, elevated salt concentrations may result in degradation, possibly due to betalain hydroxylation [59,60].

The analysis showed that the yellow pigment content of the samples decreased the most after just four days of the fermentation process, regardless of the salt content in the brine. Samples without salt showed a decrease ranging from 69 to 75% from the initial days of fermentation. In comparison, the lowest reduction in pigment content ranging from 9 to 30% was observed in samples of red beetroot fermented in brine with 5% salt. Notably, only a substantial change in yellow pigment was observed in the samples after 4 days, with no significant change during prolonged fermentation or after storage.

In the literature, it is noted that betalains exhibit higher stability at lower pH levels, particularly around pH 3–4 [59,61,62,63]. Salt can stabilize betalain protein by influencing its folded and unfolded states through interactions with charged groups. However, high salt concentrations may destabilize the pigments due to repulsive forces caused by salt ions interacting with charged groups in betalain molecules. It is important to note that betalains, particularly betacyanins, may protect against salt stress in plants by mitigating oxidative stress, thereby improving pigment stability in high salt concentrations [64,65].

The findings suggest that the ratio of red to yellow pigments is a critical factor influencing the color of the samples, as evidenced by the results of color coefficients and total color differences (dE). This statement was also mentioned by Nirmal et al. [53] and Czyżowska et al. [54].

### 2.4. Pigment Identification

Through the use of chromatography, the betalain pigments were analyzed. Results showed the existence of betanins, isobetanins, and other betacyanins, as well as yellow pigments such as vulgaxanthin-I and unidentified ones (Figure 4). The information regarding the unidentified pigments was drawn from small peaks observed in the chromatograms (Appendix A), which, in our opinion, it is also important to mention. In previous studies conducted by Czyżowska et al. [54], Nirmal et al. [53], Janiszewska-Turak et al. [1], and Choińska et al. [47], betanin, isobetanin, betanindin, and neobetanidin were identified as red pigments. In contrast, vulgaxanthin I and II and acid-betaxanthin and indicaxanthin were identified as yellow pigments. Throughout our research on the fermentation process, we observed a decrease in betanin content and an increase in isobetanin content. Following storage, we noticed the appearance of red pigments that were not identified as betanins or isobetanins and were not present in either the raw samples or the samples taken after 8 days of fermentation. In addition, isobetanin content decreased while the percentage of other red pigments increased during storage. Notably, the sample without salt exhibited the highest amount of unidentified red pigments, referred to here as “other”, and the quantity of these pigments decreased regardless of the amount of salt used during storage. The proportion of red pigments was most similar in the beetroot with 6% salt, while the proportions deviated in the samples without salt and those with 1% salt addition in the brine. The beetroot with 6% salt displayed proportions of red pigments most similar to the fresh sample, which could be related to the protective effect of the salt ions to the betacyanin molecules. In their study of fermented beetroot slices, Wiczkowski et al. [66] only detected betanin and isobetanin, instead of isobetanidin, as previously reported. Deviations were noticed in samples without salt and those with 1% salt added to the brine. This behavior, characterized by a higher proportion of isobetanin to betanin, is typically observed in juices. Researchers Wilkowska et al. [67] and Czyżowska et al. [68] noted this phenomenon in fermented beetroot juices. They also identified other betacyanins in juices not detected in fermented beetroot samples.

During the fermentation, the vulgaxanthin-I values decreased in all samples (Figure 4). Initially, yellow pigments were present in the fresh samples, with higher amounts detected in samples containing 4% salt after fermentation. However, during chromatographic analysis, these values were either much lower or undetectable in the other samples, and no yellow pigments were found in the samples after storage. The yellow pigments may transform into red pigments or experience higher degradation due to the pH sensitivity of the 60-day environment. According to the literature, betacyanins are more stable than betaxanthins. This degradation of betaxanthins could be attributed to the fermentation process, which lowered the pH and potentially led to their degradation. Betaxanthins are more sensitive to pH changes than betacyanins, as noted by Coy-Barrera [69], Fu et al. [13], and Khan [70].

Chromatograms for raw and fermented beetroot are available in the Appendix A (Appendix A), showing betacyanins and betaxanthins measured at dedicated wavelengths.

This study aimed to analyze the variations in the proportions of the four significant betalains present in red beetroot during thermal treatment. The calculations monitored the ratios of isobetanin/betanin, neobetanin/betanin, and vulgaxanthin I/betanin, proposed as an indicator by Herbach et al. [71], and the findings were documented in Table 2. It was revealed that the isobetanin/betanin ratio increased from 160 to 40% after eight days of fermentation with 0% and 6% salt addition, indicating betanin isomerization caused by a decrease in pH rather than temperature. However, it was observed that beetroot stored with 0% and 1% salt in brine experienced a decrease, suggesting that more than 8 days might be needed for the fermentation process to complete. The vulgaxanthin I/betanin ratio decreased to zero after fermentation and storage, signifying that betanin exhibited higher stability than the fresh sample. In a prior study by Choińska et al. [47], fermented beetroot with 1.5% salt addition displayed lower I/B index results and higher V/B results. The study used different LAB strains, extended fermentation time to 7 days, and a different beetroot cultivar harvested on various dates.

In this research, the N/B index could not be calculated, as neobetanin could not be identified as appropriate. However, in our case, the index related to “other betacyanin”, in which neobetanin could be present, was calculated (Table 2). After 8 days of fermentation, no other betacyanin occurred. However, after a storage period of 60 days, the presence of other betacyanin increased, leading to a change in the index from 0 to almost 1 in samples with less salt in brines. In conclusion, salt addition offers protection for betacyanin in fermented beetroot samples. Adding salt between 2% and 6% is recommended, although a trend for lower salt addition is also observed, with recommended values between 2 and 3%.

### 2.5. Correlations

In studying the impact of salt addition and fermentation duration on various physical properties of a sample, we observed specific correlations, as depicted in Figure 5. Firstly, we noted that adding salt to the brine adversely affected the pH and total acidity, showing a Pearson correlation coefficient of −0.43 and −0.40, respectively. This indicated increased salt decreased pH and total acidity compared to samples without salt. Conversely, we observed a positive correlation between salt content and the color factor b* at a coefficient of +0.64, suggesting a more yellow color with higher salt content.

Furthermore, we examined the correlation between fermentation duration and various sample properties (Figure 5). Our findings revealed a negative correlation between sample pH and fermentation duration, showcasing a coefficient of −0.66. Similarly, the red pigment content negatively correlated with fermentation duration at a coefficient of −0.70. Intriguingly, the bacterial count also negatively correlated with fermentation duration, with a coefficient of −0.80. Although this may seem counterintuitive initially, considering the anticipated increase in bacterial growth over time, we noted that storage duration may have significantly influenced these correlation coefficients. However, in testing results for those two factors without storage time, we obtained similar values (−0.75), while a decrease in eleven days occurred for all testing samples regarding the salt in the brine.

The study revealed an interesting link between pH and the physical properties of the sample. It was found that pH had a strong negative correlation with salt content, with a coefficient of −0.66. Additionally, positive correlations were observed between pH and the number of microorganisms, red pigment quantity, and sample brightness (L*). The analysis also highlighted that the color difference (dE) had the most significant impact on the sample’s brightness, with a coefficient of +0.81. However, no correlation was found between pigment content and the color coefficients L*, a*, b*.

Furthermore, the research noted that increased bacterial growth led to higher red pigment levels in the samples (Figure 5). This phenomenon was attributed to pigment release from red beetroot tissues during microbial development. Moreover, it was suggested that salt penetration into the tissues during fermentation might contribute to the easier extraction of pigments from the tissue during the study.

## 3. Materials and Methods

### 3.1. Materials

The research material was red beetroots (*Beta vulgaris* L.) cultivated and purchased in Poland. The biological material *Lactiplantibacillus plantarum* ATCC4080 was obtained from the American Type Culture Collection (ATCC, Manassas, VA, USA). Bacterial inoculum was prepared in 0.85% NaCl at a concentration of ~1 × 10^7^ CFU/mL.

### 3.2. Fermentation

The beetroots were precisely cleaned to remove any natural microorganisms. This involved washing them in tap water, peeling, and disinfecting them with a 0.5% sodium hypochlorite solution for 5 min. The cleaned beetroots were then cut into 1 cm cubes and placed in 0.5 L glass jars. Each jar was filled with a saline solution containing varying concentrations of salt (ranging from 0 to 6 percent) and a bacterial inoculum at a volume of 1% of the salt solution. Each salt concentration was tested separately, and separate jars were prepared for each day of the experiment. The fermentation process occurred in a laboratory incubator at 26 °C throughout the experiment. Each sample was analyzed on consecutive fermentation days: 0, 4, 6, 8, 11, and after 60 days of storage. Each experiment was carried out in duplicate. It was decided to use a 2% salt addition to the brine as a control condition, as this value is considered the safest and best in the literature.

### 3.3. Analytical Methods

All analytical methods were made in triplicate unless stated otherwise.

#### 3.3.1. Determination of the Number of Lactic Acid Bacteria

We followed a precise protocol to calculate the number of lactic acid bacteria present. Ten grams of beetroot sample was taken (fresh and fermented after 0, 4, 8, 14 days or 60d). Samples of beetroots were placed in sterile bags and mixed with 90 cm^3^ of physiological saline solution in a top-notch Stomacher 400 (Circulator, Seward Ltd., Worthing, UK) for 30 s. Then, 1 mL of the solution was blended with 9 mL of saline. Through this, a series of decimal dilutions were made, as well as depth cultures on MRS Agar (Biomaxima, Lublin, Poland). The plates were then incubated at a temperature of 28 °C for 48 h before the colonies were counted (ProtoCOL 3—automatic colony counting and zone measuring, Synbiosis, Frederick, MD, USA). Lastly, the results were calculated in CFU/mL and converted to log CFU/mL.

#### 3.3.2. Dry Matter

The gravimetric method was used for dry matter determination. Approximately 1 g of crushed material was weighed into prepared vessels. The weights of the empty vessel and the weighed material were recorded. The vessels were placed in a drying oven at 70 °C for 24 h. After this time, the vessels were placed in a desiccator to cool and then weighed according to information from Karwacka et al. [72].

#### 3.3.3. pH

The pH was measured using a pH meter (SevenCompact S210, Mettler–Toledo GmbH, Greifensee, Switzerland). The device was calibrated using standard buffer solutions with pH 2, 4.01, 7.00, 9.21, and 11.00. Before commencing the measurements, a verification process was conducted with a pH 7.00 buffer solution to ensure the device’s accuracy.

#### 3.3.4. Total Acidity

The total acidity was determined using the titration method as outlined in the article by Tyl and Sadler [73]. A sample was prepared by diluting a known weight of beetroot with distilled water to a volume of 150 mL and heating it for 3 min to fasten the extraction. Temperature did not exceed 50 °C. The sample was then transferred quantitatively through filter paper. A total of 25 mL of the filtrate was taken for testing and titrated with 0.1 M NaOH solution until the pH reached a value of 8.1. The amount of NaOH solution used was recorded for calculations due to the equation presented by Janiszewska-Turak et al. [74].

#### 3.3.5. Salt Content

The salt content was determined using Mohr’s method, as described by Nielsen and Nielsen [75]. The filtrate for salt determination was prepared using the same method as the total acidity method. In total, 10 cm^3^ was taken into a beaker and neutralized with 0.1 M NaOH solution. Then, 1 cm^3^ of potassium chromate solution (K_2_CrO_4_) was added and titrated with 0.1 M silver nitrate solution (AgNO_3_) until a solid brick-red color was obtained. The results were calculated from the formula:(1)NaCl content=V·M·0.05845/m·100(%)
where V—volume of AgNO_3_ solution for titration (cm^3^), M—molarity of AgNO_3_ solution (-), m—mass of the product in the titrated solution (g).

#### 3.3.6. Color

Analysis of color coefficients was made in a CR-5 spectrophotometer (Konica Minolta Sensing Inc., Osaka, Japan) in the CIE L*a*b* system. The measurement parameters were Illuminant D65, angle 2°, and calibration with white and black standards as presented by Matys et al. [76]. All measurements were made in ten repetitions.

The total color difference was calculated from the formula presented by:(2)$$dE=\sqrt{\left({\left(L^{*}-L_{F}^{*}\right)}^{2}+{\left(a^{*}-a_{F}^{*}\right)}^{2}+{\left(b^{*}-b_{F}^{*}\right)}^{2}\right)}$$
where L*/a*/b*—respectively, lightness/redness/yellowness of the sample, L_F_*/a_F_*/b_F_*—respectively, lightness/redness/yellowness of the fresh sample as a reference

#### 3.3.7. Betalain Content and Identification

Betalain was quantified using the spectrophotometric methodology delineated by [77,78]. The measurements were conducted using a Helios Gamma spectrophotometer (Thermo Spectronic Evolution 220 UV, Cambridge, UK). The pigments were extracted from the sample using a phosphate buffer with a pH of 6.5. This study used a 1 g sample of ground, fresh, and fermented beetroots mixed with 50 milliliters of buffer for 10 min. Betalain content was determined using 476, 538, and 600 nm wavelengths, with the phosphate buffer serving as the zero point.

The determination of betalain content, a red and yellow pigment, was calculated as betanin (in mg of betanin per 100 g of dry matter) and vulgaxanthin-I (in mg of vulgaxanthin-I per 100 g of dry matter).

The identification of pigments in fresh beetroot samples taken on the eighth and sixtieth days of fermentation was conducted using high-performance liquid chromatography. The methodology employed for analyzing betalains was presented by [1,78]. A mixture of 0.2% formic acid and acetonitrile was employed to extract 1 g of tissue to determine the betalains. The separation was conducted using a Waters SunFire C8 column (5 μm, 250 × 4.6 mm) with a mobile phase flow (0.2% formic acid, acetonitrile) of 1 mL/min, employing a gradient system.

Moreover, the indexes indicated that betalain changes were calculated according to the methods of Herbach et al. [71] and Choińska et al. [47]. The ratio of isobetanin to betanin provided information about the isomerization process, while the ratio of neobetanin to betanin helped determine the dehydrogenation extent. The vulgaxanthin I to betanin ratio indicated the pigment’s stability in the samples.

### 3.4. Data and Statistical Treatment

The data were analyzed using Statistica 13 software (StatSoft, Warsaw, Poland) for statistical analysis. A one-way analysis of variance with the Tukey HSD test at a significance level of α = 0.05 was conducted. Other parameters were determined using Microsoft Office 16 and the R program, including Pearson’s rank correlation analysis (*p* < 0.05) and data visualization.

## 4. Conclusions

Our study investigated the effect of different salt levels on the fermentation of red beetroot, demonstrating that salt reduction is essential to creating healthier food products. This research focused on lactic acid bacteria and revealed that bacterial growth stopped after eight days when salt levels reached 0%, 2%, and 4%, with optimal growth found at salt concentrations between 2 and 3%. Notably, salt levels fluctuated significantly during fermentation, with nearly 50% of the added salt absorbed by the beetroot tissues, mainly when lower salt concentrations were used.

The fermentation process had a negative effect on the content of red betalain pigments, particularly betanin, as well as yellow pigments, including vulgaxanthin-I. It was further shown that fermentation and storage affected the proportions of red pigments, with betacyanins proving more stable than betaxanthins.

This study observed the correlations between salt content, pH, total acidity, and various physical properties of the samples. It was found that the addition of salt had a negative effect on pH and total acidity while causing an increase in yellow color. The changing pH during fermentation was negatively correlated with the duration of the process, the amount of red pigment, and bacterial count.

In conclusion, our study provides valuable insights into the fermentation of red beetroot under different salt concentrations, shedding light on its properties. It was confirmed that the optimal salt content during fermentation leads to the preservation of probiotic bacteria, essential nutrients, and desirable color characteristics in the final product. These findings are crucial for improving the production of healthier, high-quality fermented vegetable products.

## Figures and Tables

**Figure 1 molecules-29-04803-f001:**
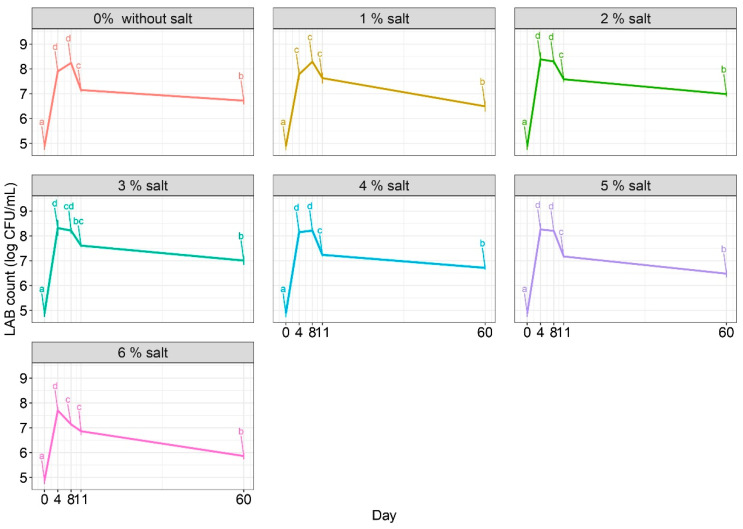
Influence of the salt addition on the LAB count. a–d Results marked with identical lowercase letters for salt addition are not considered statistically significant at α = 0.05 based on the effect of production day. For example, adding 1% salt during days 0–60.

**Figure 2 molecules-29-04803-f002:**
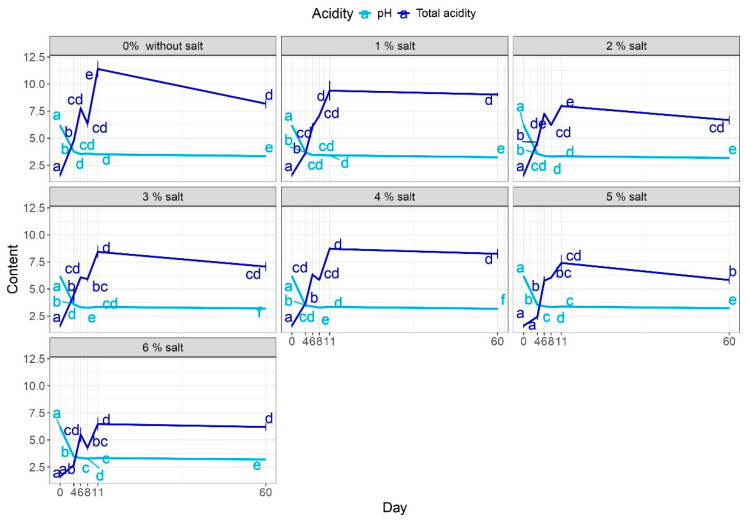
Influence of the salt addition on the pH and total acidity. a–e Results marked with identical lowercase letters for salt addition are not considered statistically significant at α = 0.05 based on the effect of production day. For example, adding 1% salt during days 0–60.

**Figure 3 molecules-29-04803-f003:**
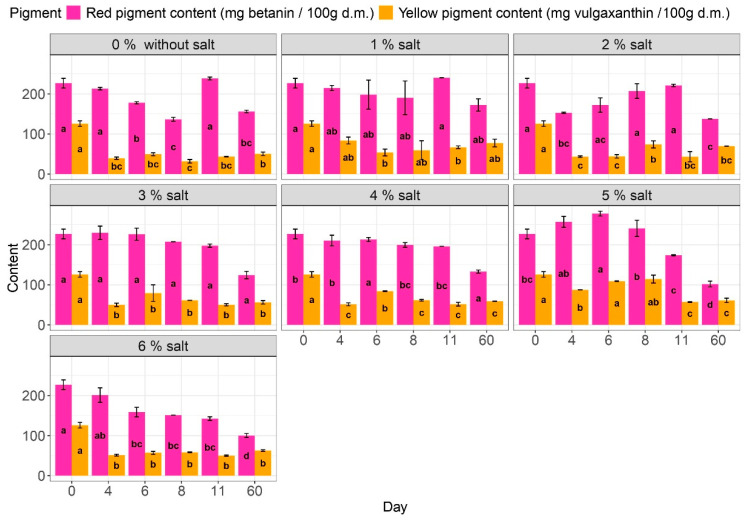
Influence of the salt addition on the pigment behavior. a–d Results marked with identical lowercase letters for salt addition are not considered statistically significant at α = 0.05 based on the effect of production day. For example, adding 1% salt during days 0–60.

**Figure 4 molecules-29-04803-f004:**
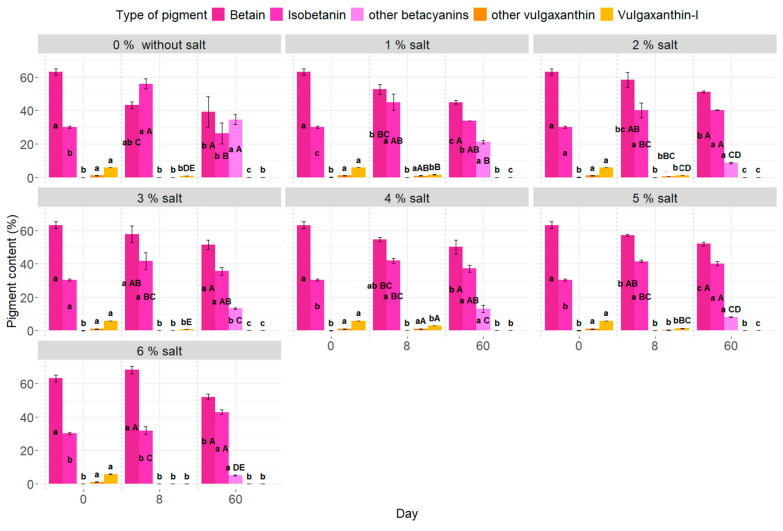
Pigment identification. a–c Results marked with identical lowercase letters for salt addition are not considered statistically significant at α = 0.05 based on the effect of production day. For example, the addition of 1% salt during days 0–60. A–E Results on a chosen fermentation day, marked with the same capital letter, do not show a statistically significant difference at α = 0.05 for salt addition. For example, salt addition from 0 to 6% on a selected day, e.g., 4.

**Figure 5 molecules-29-04803-f005:**
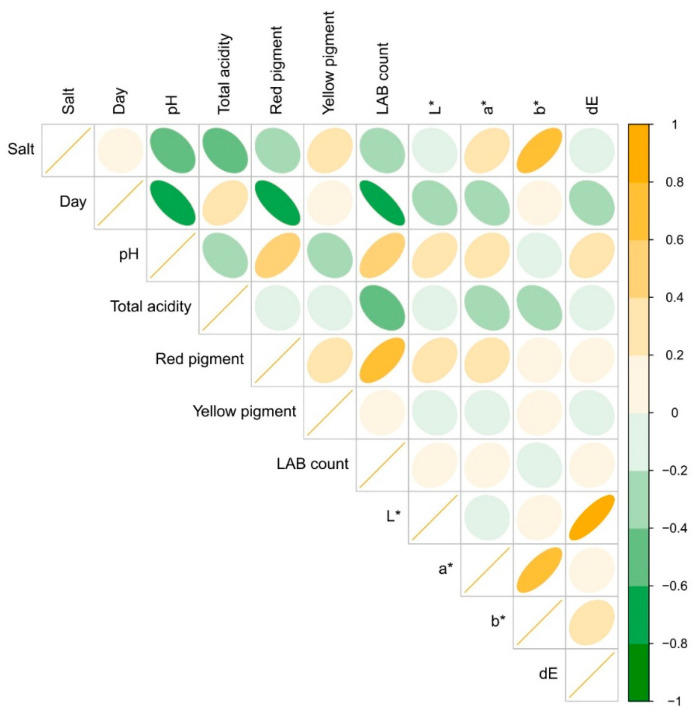
Correlation plot. In data analysis, a circular shape indicates no correlation, while more elliptical shapes signify a higher correlation. If the ellipse is positioned on the left side, it means a negative correlation, whereas if it is more to the right, it represents a positive correlation between the analyzed parameters.

**Table 1 molecules-29-04803-t001:** The impact of salt inclusion and duration of fermentation on the specific physical characteristics of beetroot fermentation.

Salt (% *w*/*v*)	Day	Dry Matter (%)	Real Salt Content in Samples (%)	Color Coefficient			dE (-)
				L*	a*	b*	
Fresh *	0	14.0 ± 1.1 ^a^	0.49 ± 0.08 ^a^	11.5 ± 0.6 ^cd^	16.3 ± 0.4 ^a^	3.6 ± 0.3 ^a^	-
0	4	7.7 ± 0.0 ^bD^	0.41 ± 0.12 ^abE^	16.8 ± 2.2 ^bA^	16.0 ± 1.6 ^aC^	−0.6 ± 0.3 ^bcD^	7.2 ± 1.0 ^cdC^
	6	6.9 ± 0.2 ^bE^	0.17 ± 0.01 ^cF^	23.4 ± 1.9 ^aA^	11.5 ± 2.6 ^bE^	−1.5 ± 0.6 ^cdE^	14.0 ± 2.1 ^aA^
	8	7.5 ± 0.0 ^bE^	0.16 ± 0.01 ^cG^	15.1 ± 4.4 ^bcA^	9.1 ± 3.4 ^bB^	−1.8 ± 0.6 ^dC^	10.8 ± 2.7 ^bA^
	11	7.3 ± 0.1 ^bE^	0.23 ± 0.05 ^bcG^	17.3 ± 4.2 ^bAB^	13.2 ± 6.0 ^abC^	−1.3 ± 0.7 ^cdD^	10.3 ± 3.5 ^bcAB^
	60	7.0 ± 0.0 ^bD^	0.20 ± 0.02 ^bcG^	11.2 ± 2.0 ^dA^	14.8 ± 2.0 ^aA^	0.1 ± 1.2 ^bC^	4.7 ± 1.3 ^dB^
Fresh *	0	14.0 ± 1.1 ^a^	0.49 ± 0.08 ^c^	11.5 ± 0.6 ^bc^	16.3 ± 0.4 ^ab^	3.6 ± 0.3 ^a^	-
1	4	7.8 ± 0.0 ^bD^	0.71 ± 0.10 ^bE^	19.7 ± 3.2 ^aA^	16.7 ± 3.0 ^abC^	−0.2 ± 0.2 ^cdCD^	9.4 ± 3.1 ^bBC^
	6	7.7 ± 0.1 ^bD^	1.16 ± 0.01 ^aE^	20.8 ± 4.2 ^aABC^	20.4 ± 6.5 ^aB^	−1.0 ± 0.3 ^dDE^	12.9 ± 1.9 ^aBC^
	8	7.8 ± 0.2 ^bDE^	0.71 ± 0.03 ^bF^	13.2 ± 2.3 ^bAB^	13.2 ± 4.4 ^bAB^	−2.0 ± 0.7 ^eC^	7.9 ± 2.4 ^bAB^
	11	7.4 ± 0.1 ^bE^	0.70 ± 0.01 ^bF^	19.3 ± 3.6 ^aA^	19.9 ± 4.2 ^aB^	1.1 ± 1.0 ^bC^	10.1 ± 2.7 ^abAB^
	60	7.4 ± 0.1 ^bD^	0.52 ± 0.02 ^cF^	8.6 ± 1.5 ^cA^	15.6 ± 1.8 ^abA^	0.4 ± 0.2 ^bcBC^	4.8 ± 0.5 ^cB^
Fresh *	0	14.0 ± 1.1 ^a^	0.49 ± 0.08 ^e^	11.5 ± 0.6 ^c^	16.3 ± 0.4 ^bc^	3.6 ± 0.3 ^a^	-
2	4	8.6 ± 0.1 ^bCD^	1.39 ± 0.10 ^aD^	19.3 ± 1.2 ^abA^	20.1 ± 1.4 ^aBC^	0.7 ± 0.1 ^bC^	9.3 ± 1.0 ^abBC^
	6	8.0 ± 0.1 ^bCD^	1.24 ± 0.00 ^abE^	18.9 ± 1.2 ^abBC^	13.2 ± 2.0 ^cdDE^	−0.6 ± 0.3 ^cD^	9.2 ± 1.6 ^abC^
	8	8.2 ± 0.0 ^bC^	1.04 ± 0.01 ^cdE^	11.9 ± 3.2 ^cABC^	15.8 ± 4.0 ^bcA^	−0.5 ± 0.4 ^cB^	6.2 ± 1.8 ^cBC^
	11	8.2 ± 0.0 ^bD^	1.10 ± 0.00 ^bcE^	17.6 ± 1.8 ^bAB^	18.5 ± 0.8 ^bcBC^	−0.6 ± 0.4 ^cD^	7.8 ± 1.8 ^bcB^
	60	8.5 ± 0.4 ^bC^	0.85 ± 0.03 ^dE^	20.6 ± 1.8 ^aBC^	11.0 ± 2.6 ^dBC^	1.1 ± 0.8 ^bBC^	10.9 ± 1.8 ^aA^
Fresh *	0	14.0 ± 1.1 ^a^	0.49 ± 0.08 ^d^	11.5 ± 0.6 ^bc^	16.3 ± 0.4 ^c^	3.6 ± 0.3 ^a^	-
3	4	9.1 ± 0.3 ^bC^	1.89 ± 0.14 ^aC^	18.4 ± 3.0 ^aA^	23.5 ± 4.1 ^abAB^	2.5 ± 1.1 ^bB^	11.1 ± 1.1 ^bAB^
	6	8.3 ± 0.0 ^bC^	1.85 ± 0.10 ^aD^	17.1 ± 3.4 ^aC^	28.1 ± 3.5 ^aA^	2.3 ± 0.7 ^bcB^	13.8 ± 1.6 ^aA^
	8	9.1 ± 0.0 ^bB^	1.70 ± 0.00 ^abD^	9.9 ± 3.0 ^bcBC^	16.1 ± 4.0 ^cA^	−0.7 ± 0.4 ^dB^	6.4 ± 1.8 ^cBC^
	11	9.0 ± 0.0 ^bC^	1.56 ± 0.02 ^bcD^	13.2 ± 2.4 ^bC^	22.1 ± 4.9 ^bAB^	1.4 ± 0.7 ^bcBC^	7.5 ± 3.0 ^cB^
	60	9.0 ± 0.0 ^bBC^	1.22 ± 0.04 ^cD^	7.1 ± 1.8 ^cA^	17.9 ± 3.9 ^bcA^	1.3 ± 0.7 ^bcBC^	6.2 ± 1.0 ^cB^
Fresh *	0	14.0 ± 1.1 ^a^	0.49 ± 0.08 ^d^	11.5 ± 0.6 ^cd^	16.3 ± 0.4 ^c^	3.6 ± 0.3 ^a^	-
4	4	10.6 ± 0.0 ^bA^	2.42 ± 0.18 ^aB^	19.0 ± 3.0 ^abA^	21.5 ± 3.6 ^aB^	2.1 ± 1.0 ^bB^	10.3 ± 1.0 ^aB^
	6	9.9 ± 0.1 ^bA^	2.10 ± 0.01 ^bC^	20.2 ± 2.2 ^aABC^	17.3 ± 0.3 ^bcBC^	1.1 ± 0.2 ^cdcC^	9.1 ± 2.1 ^abC^
	8	9.8 ± 0.1 ^bA^	1.90 ± 0.02 ^bcC^	9.8 ± 2.3 ^dBC^	10.3 ± 1.5 ^dB^	−0.9 ± 0.4 ^eB^	8.0 ± 1.1 ^bAB^
	11	9.6 ± 0.1 ^bB^	2.00 ± 0.02 ^bcC^	17.0 ± 2.6 ^abAB^	21.0 ± 5.1 ^abB^	1.4 ± 0.7 ^bcBC^	9.2 ± 1.4 ^abAB^
	60	9.4 ± 0.2 ^bB^	1.77 ± 0.04 ^cC^	16.2 ± 3.6 ^bcC^	9.0 ± 1.4 ^dC^	0.7 ± 0.6 ^dBC^	9.7 ± 1.2 ^abA^
Fresh *	0	14.0 ± 1.1 ^a^	0.49 ± 0.08 ^e^	11.5 ± 0.6 ^c^	16.3 ± 0.4 ^c^	3.6 ± 0.3 ^b^	-
5	4	8.5 ± 0.1 ^bcBC^	2.63 ± 0.06 ^AB^	20.1 ± 2.4 ^aA^	26.1 ± 3.0 ^aA^	4.5 ± 0.9 ^aA^	13.4 ± 2.4 ^aA^
	6	9.0 ± 0.0 ^bcB^	2.59 ± 0.03 ^abB^	21.0 ± 3.0 ^aAB^	14.8 ± 3.1 ^cC^	1.2 ± 0.7 ^dCD^	10.5 ± 2.5 ^bBC^
	8	8.0 ± 0.1 ^cCD^	2.45 ± 0.03 ^bcB^	8.9 ± 0.5 ^cC^	17.0 ± 1.7 ^cA^	0.3 ± 0.2 ^eA^	4.5 ± 0.5 ^cC^
	11	10.1 ± 0.1 ^bA^	2.38 ± 0.00 ^cB^	16.8 ± 2.7 ^bABC^	21.7 ± 2.4 ^bB^	2.2 ± 0.1 ^cAB^	8.2 ± 1.9 ^bB^
	60	9.8 ± 0.1 ^bAB^	2.11 ± 0.02 ^dB^	18.7 ± 0.8 ^abBC^	10.8 ± 1.5 ^cBC^	1.8 ± 0.5 ^cdB^	9.4 ± 0.7 ^bA^
Fresh *	0	14.0 ± 1.1 ^a^	0.49 ± 0.08 ^d^	11.5 ± 0.6 ^d^	16.3 ± 0.4 ^c^	3.6 ± 0.3 ^abc^	-
6	4	10.7 ± 0.1 ^bB^	3.00 ± 0.04 ^bA^	19.4 ± 2.2 ^abA^	22.9 ± 3.4 ^bAB^	2.7 ± 0.7 ^cB^	10.9 ± 2.0 ^bcB^
	6	9.3 ± 0.0 ^bB^	3.20 ± 0.02 ^abA^	16.9 ± 0.5 ^bcC^	29.8 ± 0.5 ^aA^	3.9 ± 0.2 ^abA^	14.5 ± 0.4 ^aA^
	8	9.8 ± 0.1 ^bA^	3.20 ± 0.08 ^aA^	15.8 ± 3.9 ^cA^	10.4 ± 3.3 ^dB^	0.3 ± 0.2 ^dA^	8.7 ± 3.8 ^cAB^
	11	10.1 ± 0.1 ^bA^	3.29 ± 0.02 ^aA^	14.9 ± 1.0 ^cBC^	27.8 ± 3.0 ^aA^	2.8 ± 0.8 ^bcA^	12.1 ± 2.7 ^abA^
	60	10.4 ± 0.1 ^bA^	2.59 ± 0.06 ^cA^	20.5 ± 2.3 ^aAB^	14.8 ± 5.3 ^cdAB^	4.4 ± 2.6 ^aA^	10.7 ± 1.4 ^abcA^

* The fresh sample refers to the beetroot sample before fermentation. Information for beetroot values is placed multiple times in Table 1 for easier statistical analysis (homogeneous groups). a–e—results marked with the same lowercase letters separately for salt addition are not statistically significantly different at α = 0.05 (effect: production day), e.g., 1% of salt and days 0–60. A–G—results on the selected day of fermentation marked with the same capital letter are not statistically significantly different at α = 0.05 (effect: salt addition), e.g., day 4 for salt addition from 0 to 6%.

**Table 2 molecules-29-04803-t002:** Indexes for pigment changes (calculated based on Choińska et al. 2022 and Herbach et al. 2006 [47,71]) I/B—isobetanin/betanin—isomerization index; V/B—vulgaxanthin-I/betanin—stability index; O/B—other betacyanin/betanin—changes in samples in betacyanin pigments.

Day	Salt (%)	Indexes		
		I/B	V/B	O/B
0	0	0.5	0.1	0.0
8	0	1.3	0.0	0.0
	1	0.9	0.0	0.0
	2	0.7	0.0	0.0
	3	0.7	0.0	0.0
	4	0.8	0.1	0.0
	5	0.7	0.0	0.0
	6	0.5	0.0	0.0
60	0	0.7	0.0	0.9
	1	0.8	0.0	0.5
	2	0.8	0.0	0.2
	3	0.7	0.0	0.3
	4	0.7	0.0	0.3
	5	0.8	0.0	0.2
	6	0.8	0.0	0.1

## Data Availability

The original contributions presented in the study are included in the article; further inquiries can be directed to the corresponding author.

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
