# Peer review of "Studying the Influence of Salt Concentrations on Betalain and Selected Physical and Chemical Properties in the Lactic Acid Fermentation Process of Red Beetroot"

_molecules, 2024, doi:10.3390/molecules29204803_

Round 1
Reviewer 1 Report
Comments and Suggestions for Authors
The scientific quality of the work presented in the manuscript is generally satisfactory, but I cannot recommend publication in its current form due to several areas needing improvement. Here are the main drawbacks:
- Abstract: The abstract highlights the importance of salt concentration but lacks specific quantitative results. Including these details would help clarify the findings for readers.
- Introduction: The introduction references multiple studies but does not provide enough context or comparison. It would be beneficial to include more specific comparisons to previous research, particularly regarding how salt concentration impacts bacterial growth. Additionally, the study's objective could be articulated more clearly to frame the research questions being explored.
- Materials and Methods: The manuscript does not specify the control conditions used during fermentation. This information is crucial for understanding the experimental setup.
- Results: Some figures would benefit from improved resolution to enhance clarity and readability.
- Discussion: The discussion section lacks adequate comparisons with existing literature. It should contextualize findings within the broader field to highlight their significance.
- Conclusions: The conclusions should succinctly summarize the key findings and their implications without introducing any new data or interpretations.
- General Comments: There are minor grammatical errors and instances of awkward phrasing throughout the manuscript. A thorough proofreading is recommended to improve the overall readability.
Comments on the Quality of English Language
Minor editing of English language required.
Author Response
Comments and Suggestions for Authors |
Answers |
The scientific quality of the work presented in the manuscript is generally satisfactory, but I cannot recommend publication in its current form due to several areas needing improvement. Here are the main drawbacks: |
Thank you for your valuable work as a reviewer. Your comments are greatly appreciated and can help improve our work. |
Abstract: The abstract highlights the importance of salt concentration but lacks specific quantitative results. Including these details would help clarify the findings for readers. |
Result information has been added to the abstract. |
Introduction: The introduction references multiple studies but does not provide enough context or comparison. It would be beneficial to include more specific comparisons to previous research, particularly regarding how salt concentration impacts bacterial growth. Additionally, the study's objective could be articulated more clearly to frame the research questions being explored. |
The information has been included. We trust that it provides sufficient clarity on the process.
|
Materials and Methods: The manuscript does not specify the control conditions used during fermentation. This information is crucial for understanding the experimental setup. |
We decided to use a 2% salt addition to the brine as a control condition, as this value is considered the safest and best in the literature. This information have been added to the manuscript. |
Results: Some figures would benefit from improved resolution to enhance clarity and readability. |
The figures have been updated to improve clarity. |
Discussion: The discussion section lacks adequate comparisons with existing literature. It should contextualize findings within the broader field to highlight their significance. |
Discussion part has been changed following the reviewer's comments. |
Conclusions: The conclusions should succinctly summarize the key findings and their implications without introducing any new data or interpretations. |
The conclusion has been shortened and clarified. |
General Comments: There are minor grammatical errors and instances of awkward phrasing throughout the manuscript. A thorough proofreading is recommended to improve the overall readability. |
Thank you for your comment. We have also checked the English after making the correction. |

Reviewer 2 Report
Comments and Suggestions for Authors
The study provides an extensive evaluation of how different salt concentrations affect various parameters during the fermentation of red beetroot, including pH, dry matter content, total acidity, pigment content, and lactic acid bacteria counts.
Weak points:
-
Abstract Improvement: The abstract should clearly present the most important results with specific data. Including quantitative findings will enhance the abstract's impact and provide a quick summary of the study’s significance.
-
Detailed Pigment Analysis: The statement "Through the use of chromatography, the betalain pigments were analysed. The results indicated the presence of betanins, isobetanins, and other betacyanins, as well as yellow pigments including vulgxanthin-I" is vague. It is essential to clarify what "other betacyanins" refers to and provide detailed results for each pigment identified. The study should include specific quantitative data on each pigment and present real chromatograms in the supplementary material to support the findings.
-
Extraction Temperature: The study does not specify the temperature used during the extraction process. This information is crucial as temperature can significantly affect pigment stability and overall extraction efficiency.
-
Fermentation Challenges with Low Salt Levels: The study does not address potential issues with fermentation at low salt levels, such as the risk of spoilage or off-flavors. It is important to discuss whether any undesirable processes, such as rotting, were observed during the fermentation with minimal salt content.
-
Limited Exploration of Salt Mechanisms: While the study effectively outlines the effects of salt concentrations on fermentation, it does not deeply explore the underlying biochemical mechanisms of how salt influences lactic acid bacteria growth or betalain stability. Including a discussion on the biochemical interactions between salt and bacterial growth would strengthen the paper.
-
Conciseness of Conclusions: The conclusions section is lengthy and should be revised to be clearer and more concise. It should succinctly summarize the key findings and their implications, avoiding repetition and excessive detail.
Author Response
Comments and Suggestions for Authors
|
Answers |
The study provides an extensive evaluation of how different salt concentrations affect various parameters during the fermentation of red beetroot, including pH, dry matter content, total acidity, pigment content, and lactic acid bacteria counts. |
Thank you for your valuable work as a reviewer. Your comments are greatly appreciated and can help improve our work.
|
Abstract Improvement: The abstract should clearly present the most important results with specific data. Including quantitative findings will enhance the abstract's impact and provide a quick summary of the study’s significance. |
Result information has been added to the abstract.
|
Detailed Pigment Analysis: The statement "Through the use of chromatography, the betalain pigments were analysed. The results indicated the presence of betanins, isobetanins, and other betacyanins, as well as yellow pigments including vulgxanthin-I" is vague. It is essential to clarify what "other betacyanins" refers to and provide detailed results for each pigment identified. The study should include specific quantitative data on each pigment and present real chromatograms in the supplementary material to support the findings. |
Thank you for your valuable comment. I have added supplementary material with chromatograms as you suggested. I want to clarify that the peaks identified as "other betaxanthins" or "other betacyanins" were several low-intensity peaks, but their qualitative identification was not certain. In the samples after fermentation, the amount of other single compounds was trace, so they were expressed as a sum. According to the literature, as a result of the fermentation processes in beetroots, compounds from the betacyanin group, such as neobetanin, could appear; however, the identification is not unequivocal. Betaxanthins were quantified at 480 nm, and betanin and isobetanin (as betacyanins) were quantified at 538 nm due to much bigger peaks occurring at this wavelength. |
figures from suplemantary fiele here | |
Extraction Temperature: The study does not specify the temperature used during the extraction process. This information is crucial as temperature can significantly affect pigment stability and overall extraction efficiency. |
We have added this information to the text. The extraction for all measurements was at room temperature (20-22oC). This temperature is not high for pigments and sensitive ingredients. In total acidity measurement the temperature did not exceed 50oC. |
Fermentation Challenges with Low Salt Levels: The study does not address potential issues with fermentation at low salt levels, such as the risk of spoilage or off-flavors. It is important to discuss whether any undesirable processes, such as rotting, were observed during the fermentation with minimal salt content. |
The information has been included. We trust that it provides sufficient clarity on the process. “In all the samples, no signs of spoilage were detected. There was no visible presence of yeasts and molds. Additionally, the taste and smell were typical for fermented vegetables. A pronounced salty taste was observed when higher salt levels were added to the brine no changes in the structure and texture of the beetroot cubes were observed, for our other studies on yellow beetroot fermentation, similar observations for structure/texture were made [39].” |
Limited Exploration of Salt Mechanisms: While the study effectively outlines the effects of salt concentrations on fermentation, it does not deeply explore the underlying biochemical mechanisms of how salt influences lactic acid bacteria growth or betalain stability. Including a discussion on the biochemical interactions between salt and bacterial growth would strengthen the paper. |
The information has been included. We trust that it provides sufficient clarity on the process.
|
Conciseness of Conclusions: The conclusions section is lengthy and should be revised to be clearer and more concise. It should succinctly summarize the key findings and their implications, avoiding repetition and excessive detail. |
The conclusion has been shortened and clarified.
|

Round 2
Reviewer 2 Report
Comments and Suggestions for Authors
The article has been significantly revised; however, despite indicating that some data are presented in the supplementary materials, this attachment does not contain them under that name.
Author Response
Reviewer
The article has been significantly revised; however, despite indicating that some data are presented in the supplementary materials, this attachment does not contain them under that name.
Authors response.
Dear Editor,
Thank you for the next revision.
We have checked the names and citations in the text and the supplementary file; they are correctly cited and named. The figure titles are the same in both the manuscript and the supplementary file.
However, if you are asking for the web address that is placed in the manuscript, we have left the one that was in the template, as we did not know that we had to create it. In our opinion, the editorial office will do that after manuscript acceptance.
With best regards
Emilia Janiszewska-Turak
(on behalf of all authors)
